# The Reciprocal Relationship between Frailty and Depressive Symptoms among Older Adults in Rural China: A Cross-Lag Analysis

**DOI:** 10.3390/healthcare9050593

**Published:** 2021-05-17

**Authors:** Xuehui Wang, Kaijun Shen

**Affiliations:** 1Center for Population and Development Policy Studies, School of Social Development and Public Policy, Fudan University, Shanghai 200433, China; wxhui@fudan.edu.cn; 2School of Social Development and Public Policy, Fudan University, Shanghai 200433, China

**Keywords:** frailty, depressive symptoms, cross-lag model, older adults

## Abstract

(1) Objective: This study aimed to investigate the reciprocal relationship between frailty and depressive symptoms using longitudinal data among older adults in China. (2) Methods: Data derived from 2014 and 2017 waves of a longitudinal study of 1367 older adults aged 70–84 years, living in rural areas of Jiangsu Province, China. Cross-lagged panel model and a multiple group model were used to examine the temporal effect of frailty on depressive symptoms and vice versa. (3) Results: Frailty was associated with subsequent increase in depressive symptoms, such that participants with higher levels of frailty increase the risks of depressive symptoms (b = 0.090, *p* < 0.01). Depressive symptoms were significant predictors of increased frailty (b = −0.262, *p* <0.001). However, older men and older women had no significant differences in the reciprocal relationship between frailty and depressive symptoms. (4) Conclusions: In conclusion, we find a significant bi-directional relationship between frailty and depressive symptoms. This finding confirms the dyadic model of frailty and depression. Implications for interventions and policy to help frail and depressive older adults are discussed.

## 1. Introduction

Frailty and depressive symptoms are two common health issues among older adults. According to previous studies, the prevalence of frailty varies from 4.9% to 27.3% in community-dwelling adults aged 65 and over [1], the prevalence of depressive symptoms in community samples of adults aged 65 and older ranges from 1% to 5% in most large-scale epidemiological investigations [2]. Both may lead to adverse health outcomes, such as poor health-related quality of life, increased use of healthcare services, disabilities, falls, institutionalization and mortality [3,4,5,6]. Among older adults, frailty often co-occurs with the deterioration in mental health, suggesting a close relationship between the two [7,8,9]. For instance, in multiple cross-sectional studies, frail older adults had higher odds of developing depressive symptoms compared with those without frailty, and older adults with depressive symptoms had a higher risk of being frail than those without depressive symptoms [10,11,12,13,14,15].

Among older adults, the parallels between frailty and depression leads to the hypothesis of comorbidity [16]. So far, many studies have pointed out that there is a large degree of overlap between frailty and depression assessment criteria, such as unconscious weight loss, and experience of exhaustion, but they also have their own unique symptoms. The connection between the two cannot be explained simply by the overlap of symptoms. Frailty and depression have their own characteristics. Although many studies have revealed that frailty and depression are highly correlated, there is no direct evidence that the two are causal. A prospective cohort study in Italy with a follow-up of 4.4 years found that there is no evidence that pre-frailty per se predisposes to the onset of depression, and improvement in frailty status is not associated with the risk of incident depression [17]. Similarly, a prospective cohort study from England found that frailty and pre-frailty did not predict the onset of depression among older community dwellers, when accounting for potential confounders, only slow gait speed may predict depression in older adults [18]. There are several reasons that may explain the high levels of comorbidity between frailty and depression. One is the overlap in some areas of diagnostic criteria, such as unintentional weight loss, making it difficult to distinguish between individuals. Another factor is that frailty and depression have some common etiology [19].

The above studies show that frailty and depression are highly overlapping and have their own characteristics, and there is no sufficient evidence to support potential two-way effects. Frailty or symptoms of frailty may have a certain predictive effect on the occurrence of depression, and intervention in frailty may prevent depressive symptoms. However, the influence of frailty level on the prevalence of depression is controversial, and there is insufficient evidence to prove that antidepressant treatment is beneficial to depressive older adults to avoid or delay frailty [5,6,7].

According to existing studies, most research objects are mainly elderly people in European and American countries [9]. Due to the large differences in social and cultural backgrounds and medical service systems between different countries, it is necessary to include more data from other countries to examine the correlation between frailty and depression according to the demographic characteristics of specific regions, in order to assess and identify frailty and depression in practice, and timely intervention measures to prevent or reduce the occurrence and development of the disease, to avoid the worse results. Moreover, understanding the reciprocal relationship between frailty and depressive symptoms will help health professionals identify whether and how frailty leads to a decline in mental health. Health professionals will also benefit from recognizing the influence of mental health deterioration on frailty.

In general, poor physical health can lead to an increased risk of mental health problems. In a similar vein, poor mental health can negatively impact physical health. The two may have a causal relationship with each other. Thus, given the aforementioned knowledge gaps in the prospective relationships between frailty and depression, this study aimed to investigate the bi-directional relationship between frailty and depression in a cohort of older adults in rural China. Understanding the reciprocal relationship between frailty and depressive symptoms is of great value for the screening and prevention of frailty and depression in older adults.

In addition, compared with older men, older women are more likely to be identified as frail and tend to have higher risk of depressive symptoms [20,21]. However, almost no study has investigated the gender differences in the relationship between frailty and depressive symptoms. To address this gap, we aimed to explore the gender differences in the bi-directional relationship between frailty and depressive symptoms in community-dwelling elders aged 70 to 84 years in rural China.

Building upon previous studies, we developed the following hypothesis:

**Hypothesis** **1.**
*The higher level of frailty may predict a higher level of depressive symptoms.*


**Hypothesis** **2.**
*The higher level of depressive symptoms predicts a higher level of frailty.*


## 2. Materials and Methods

### 2.1. Study Design and Participants

We used data from the ageing arm of the Rugao Longevity and Ageing Study (RuLAS), a community-based survey conducted in Rugao, Jiangsu Province, China [22]. This study enrolled 1788 older adults aged 70–84 years at baseline (November to December of 2014) from 31 rural communities of Jiang’an Township, Rugao City. Two follow-up surveys were conducted 1.5 years and 3 years after the baseline. The study was approved by the Institutional Review Board of the School of Life Sciences, Fudan University. Written informed consent was obtained from each participant prior to the study. For the purpose of this study, we use the data wave1 and wave3 over a three-year period. Deleting the participants who died before the third wave (*N* = 149), and who have missing values on the key variables (*N* = 263), the analysis sample includes 1376 older adults, with 622 older men and 754 older women, respectively.

### 2.2. Assessment of Frailty

Frailty was assessed by frailty index in the study. According to Rockwood and Mitnitski, a frailty index (FI) counts deficits in health which were defined as symptoms, signs, disability and diseases [23]. We used 42 health deficits including symptoms, comorbidities, activities of daily living (basic and instrumental) and cognitive function to construct a FI. Symptoms include hearing impairment, vision impairment, dizziness, headache, bodily pain, weight loss, feel exhaustion, speech problem, taking 5 or more medications, quality of sleep, urinary incontinence, fecal incontinence, self-rated health, and health change. Comorbidities include hypertension, diabetes, coronary heart disease, cerebrovascular disease, chronic lung disease, arthritis, osteoporosis, thyroid disease, eye disease, arrhythmia, fractures, Parkinson’s disease, and cancer. Activities of daily living include bathing, dressing, eating, indoor transfer, toileting, incontinence, preparing meals, doing housework, taking a bus, shopping for necessities, managing money, washing clothes by oneself, taking medication, and using atelephone. Cognitive function was measured using the Mini-Mental Status Examination (MMSE) in two waves. The detailed description of constructing FI was provided in a previous study [24]. In short, each deficit variable was dichotomized to 0 (0 as no deficit) and 1 (1 as with deficit) to represent the severity of the deficit. We summed the scores of the deficits in each dimension to produce four continuous variables of Frailty in the four dimensions, including symptoms, comorbidities, activities of daily living and cognitive function, ranging from 0.02 to 0.71 in 2014 and from 0.01 to 0.69 in 2017.

### 2.3. Assessment of Depressive Symptoms

Depressive symptoms were assessed by using the GDS-15 (Geriatric Depression Scale) [25]. Respondents were asked to give their answers to 15 items. For each item, participants needed to answer yes or no (1 = yes, 0 = no). We recoded the value of four positive items and summed the 15 items to present the levels of depressive symptoms, ranging from 0 to 15. Higher scores indicate higher levels of depressive symptoms. The internal consistency of GDS-15 was good in both waves (Cronbach’s α = 0.72 in 2014 and 0.76 in 2017).

### 2.4. Statistical Analysis

To examine the longitudinal associations between frailty and depressive symptoms, the cross-lagged panel model was used. The cross-lagged panel model is a type of structural model, which is suitable for the context where two or more variables are observed at two or more time points [26]. The aim of such a model was to explore the association between different variables at different time points. A multiple group model was also used to explore gender differences in the association between frailty and depressive symptoms.

We used SPSS 21 for descriptive statistics and used the structural equation modeling software Amos 24 for the analysis of cross-lagged panel model [27]. We conducted the cross-lagged model analysis as the following. First, we established measurement models of frailty from wave 1 and wave 3 by using confirmatory factor analysis separately. Second, reciprocal causality between frailty and depressive symptoms was examined by using a two-wave cross-lagged model. Third, gender differences of the bi-directional relationship between frailty and depressive symptoms were examined using a multiple group model.

Measurement models of Frailty were established during the 2014 and 2017 waves before testing of the two-wave cross-lagged model. Confirmatory factor analysis was used to examine the relationships between the factor indicators and the corresponding factors, including factor loadings, factor variance, and correlations. Maximum likelihood estimation was the default estimation technique. The latent construct of frailty had four indicators: symptoms, comorbidities, ADL_IADL, cognitive function. The standardized factor loadings of the four indicators of frailty in 2014 ranged from 0.236 to 0.768 and frailty in 2017 ranged from 0.164 to 0.654, and were all statistically significant at the 0.001 level.

## 3. Results

### 3.1. Socio-Demographic Characteristics of Participants in Baseline

Table 1 shows demographic and health characteristics of participants at baseline. The mean age of participants was 75.1 years (Range = 70–84 years, SD = 3.802). There were more females than males (54.8% and 45.2%, respectively). Nearly two thirds of participants were married at baseline. More than half of participants were illiterate, and more than 90% of participants were farmers.

### 3.2. Correlations between Depressive Symptoms and Frailty

Correlations between depressive symptoms and frailty across the two waves were provided in Table 2. The associations between each variable during the two time points were statistically significant except comorbidities and cognitive function, showing that all variables were stable in the two waves. In addition, the associations of all variables were similar between 2014 and 2017, indicating that the associations were stable. Meanwhile, we also made several linear regression analyses to examine the existence of multicollinearity in the data. The results showed that all variance inflation factor estimates were less than 10 and tolerance estimates were greater than 0.1. Thus, there is no multicollinearity problem between variables.

### 3.3. Two-Wave Cross-Lagged Panel Model

We tested the reciprocal relationship between frailty and depressive symptoms by using the cross-lagged panel model. The results of the fit indices indicated adequate model fit, X2 (25, *N* = 1376) = 11.605, *p* = 0.237, RMSEA= 0.015, NFI = 0.995, TLI = 0.995, CFI = 0.999. Standardized factor loadings of frailty ranged from 0.298 to 0.796 and were all statistically significant at the 0.001 level. After controlling sociodemographic variables, such as age, gender, marital status and education, the paths between the two waves for frailty (β = 0.491, *p*< 0.001) and the two waves for depressive symptoms (β = 0.141, *p* < 0.001) were both statistically significant at the 0.001 level. Meanwhile, frailty was significantly associated with depressive symptoms at each of the two waves; the coefficient was 0.383 (*p* < 0.001), depressive symptoms 0.252 (*p* < 0.001) individually.

Among the cross-lagged correlations of frailty and depressive symptoms, the results showed that 2014 frailty was a significant predictor of 2017 depressive symptoms (β = 0.090, *p* < 0.01), and 2014 depression was a significant predictor of 2017 frailty (β = 0.262, *p* < 0.001). The results are summarized in Figure 1.

### 3.4. Gender Effects

The multiple group model showed a good fit to the data (all RMSEA < 0.05). In measurement weights model (*p* = 0.001), which indicated significance in measurement model weights (factor loading) between male and female participants. In pairwise parameter comparisons, the critical ratios of comorbidities in 2017 and cognitive function in 2017 among older men and older women were 1.964 and −2.998, respectively, which means that the impact of comorbidities and cognitive function on frailty at wave 3 was significantly different between males and females. In structural weights model, all the absolute values of critical ratios among frailty in 2014, frailty in 2017, depressive symptoms in 2014 and depressive symptoms in 2017 were less than 1.96, which indicated that there was no significant gender difference in the cross-lagged group model.

## 4. Discussion

Frailty and depression both cause suffering and burden for individuals, families and society. To develop strategies targeted to lower the burden and improve life quality of older adults, a detailed examination of the relationship between physical health and mental health is warranted. The aim of the present study was to investigate the reciprocal relationship between frailty and depressive symptoms. The findings support the proposed dyadic model of frailty and depressive symptoms among older adults, indicating a significant reciprocal relationship between frailty and depression. This study contributes to the literature in that it explored to which extent the relationship between frailty and depression can be explained by reciprocal influence among older adults in rural China. The findings could be helpful for developing policies and interventions to address the needs of older adults with frailty or depression or both.

Two hypotheses were supported by the analysis results. Consistent with previous studies, we found that frailty predicts the level of depressive symptom which means that those with a higher degree of frailty showed a higher level of depressive symptoms in the following wave, which supported Hypothesis 1. Depressive symptoms at baseline predict the level of frailty at follow-up, which supported Hypothesis 2. Therefore, the proposed dyadic model was confirmed by the findings in this study.

The first hypothesis that frailty is a predictor of depressive symptoms (H1) was supported. This finding is consistent with previous studies conducted in other countries. For example, two studies found that frailty was a significant predictor of depressive symptoms among older adults in the U.S. and Singapore [11,28]. This may be because older adults in frailty are more likely to have poorer physical function, more limited social activities, and smaller social support networks. In addition, frail older adults usually face physiological dysfunction which may affect depression, such as changes in the digestive system or immune system, causing older adults to feel depressed [11,29]. One explanation of the result is that dysfunction in basic systems of the body might trigger a cascade of processes resulting in more depressive symptoms [30]. The second hypothesis that depressive symptoms are a predictor of frailty (H2) was also supported. The finding is consistent with one previous study in Mexican American older adults that an association between negative affect scores on the CES-D subscale at baseline and a risk of frailty at the 10-year follow-up was found in a longitudinal study [31]. This may because psychosocial factors, such as specific personality traits (e.g., neuroticism) or loneliness, might cause higher levels of depression [32].

In addition, although older females had higher risk of frailty and having depressive symptoms [18], the results showed that there was no significant gender difference in the causal relationship between frailty and depressive symptoms. This result was consistent with a study from America [33]. Presumably, males and females may have different impact on the relationship between frailty and depressive symptoms because male and female older adults had strong heterogeneity in health condition with different characteristics. However, our study did not find this relationship.

The findings of this study indicate the necessity of developing preventions and interventions to support older adults in rural China. It is necessary to carry out early screening and diagnosis of frailty and depressive symptoms among community-dwelling older adults, which helps to alleviate their deterioration. Meeting the different needs of frail and depressed older adults requires the cooperation of multiple departments, such as local government, communities, families, social organizations, health service sectors and the development of new interventions. It is also necessary to advocate for older adults to pay attention to the risk of frailty and depression. In China, frailty and depression of older adults have not attracted enough attention. Given the rapid development population aging and increased number of frail older adults, a nationwide policy supporting the diagnosis of early frailty and depression of older adults is called for, which will increase healthy life expectancy and achieve healthy aging in China.

This study has some limitations. First, participants in the study live in rural areas, and thus are not representative of the entire older population in China. Usually, older adults in rural areas have lower social economic status and higher labor participation, which may increase the level of frailty and depressive symptoms [34]. Second, the age of participants in this study may be relatively older, which may affect the analysis results to a certain extent. Considering younger older adults are less likely to have frailty in general, the older subjects will not affect the validity of the study. In addition, we just chose respondents from both the 2014 and 2017 waves of the survey in the analysis. Excluding the death sample and missing data sample may lead to selection bias, although sensitive analysis results showed that there was no significant difference between the two samples.

However, there were several strengths in this study. First is the examination of the reciprocal relationship between frailty and depressive symptoms. This study also investigated the gender differences in the association between frailty and depressive symptoms. Finally, the study used more suitable and standardized instruments to measure frailty and depressive symptoms.

## 5. Conclusions

In conclusion, the present study found that frailty was a significant predictor for depressive symptoms. However, a depressive symptom was not a significant predictor of frailty. Therefore, this study found that there was no reciprocal relationship between frailty and depression. Although we examined the gender difference in the causal relationship between frailty and depressive symptoms among male and female participants, the results have not shown a significant gender differences in the association. Future research should examine whether a shared pathology exists for the causal relationship between frailty and depression. Such research might also consider some moderators in the relationship between the two, such as physical activity, lifestyle, medication.

Both frailty and depressive symptoms have adverse effects on the health statuses of older adults. The results of this study have shown that older adults with frailty were more likely to have depressive symptoms, which would lead to worse health conditions. Therefore, it is necessary to assess depressive symptoms of older adults who are with frailty. Good physical health is important for mental health and quality of life among older adults. The results of this study also demonstrate that there may exist different mechanisms in the causal relationship between frailty and depressive symptoms. However, it is highly likely that frailty could predict depressive symptoms among older adults in rural areas of China. Therefore, relevant aging policies in rural China should pay more attention to the improvement of physical health of older adults, such as providing routine medical examination for older adults aged 60 years and over. In addition, we also need to encourage family members to keep in touch with older adults in order to maintain their mental health and prevent the occurrence of depression at an early stage.

## Figures and Tables

**Figure 1 healthcare-09-00593-f001:**
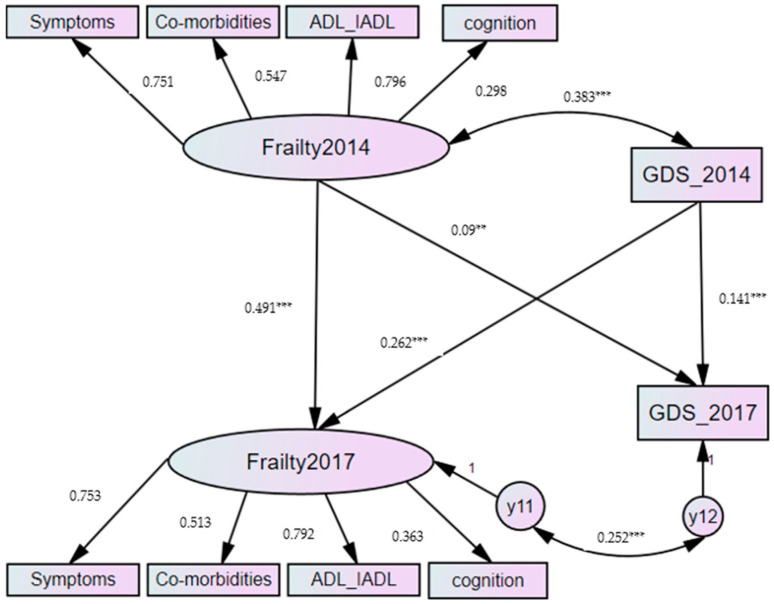
Results of the cross-lagged panel model. ** *p* < 0.01; *** *p* < 0.001.

**Table 1 healthcare-09-00593-t001:** Demographic and health characteristics of participants at baseline (N = 1376).

	*N* (%)	Mean (SD)
Age		75.1 (3.802)
70–74	695 (50.5)	
75–79	466(33.9)	
80–84	215 (15.6)	
Gender		
Male	622 (45.2)	
Female	754 (54.8)	
Marital status at baseline		
Currently Married	930 (68.2)	
Others	434 (31.8)	
Educational level at baseline		
Illiterate	735 (54.4)	
Literate	617 (45.6)	
Occupation		
Farmer	1225 (90.4)	
Others	130 (9.6)	
Frailty Index		
≤0.1	193 (14.0)	
0.1–0.21	716 (52.0)	
>0.21	467 (33.9)	

**Table 2 healthcare-09-00593-t002:** Bivariate correlations between depressive symptoms and frailty among participants (*n* = 1376).

Indicators	M	SD	1	2	3	4	5	6	7	8	9	10
1. Depression 2014	2.46	2.278	1.000									
2. Symptoms 2014	4.102	1.773	0.344 ***	1.000								
3. Co-morbidities 2014	1.340	1.097	0.148 ***	0.277 ***	1.000							
4. ADL_IADL 2014	1.506	2.106	0.309 ***	0.331 ***	0.205 ***	1.000						
5. Cognitive function 2014	0.50	0.247	0.575 ***	0.177 ***	0.114 ***	0.174 ***	1.000					
6 Depression 2017	2.00	2.302	0.141 ***	0.084 ***	0.057 *	0.059 *	0.086 **	1.000				
7. Symptoms 2017	3.902	1.703	0.256 ***	0.390 ***	0.166 ***	0.223 ***	0.150 ***	0.315 ***	1.000			
8. Co-morbidities 2017	1.344	1.125	0.163 ***	0.219 ***	0.423 ***	0.126 ***	0.108 ***	0.123 ***	0.318 ***	1.000		
9. ADL_IADL 2017	1.690	2.104	0.161 ***	0.211 ***	0.105 ***	0.433 ***	0.054	0.196 ***	0.318 ***	0.151 ***	1.000	
10. Cognitive function 2017	0.54	0.251	0.135 ***	0.129 ***	0.049	0.035	0.094 ***	0.392 ***	0.287 ***	0.118	0.204 ***	1.000

M, Mean; SD, Standard deviation; GDS, Geriatric Depression Scale; SP, Symptoms; CM, Co-morbidities; CF, Cognitive function * *p* < 0.05; ** *p* < 0.01; *** *p* < 0.001 (all two-tailed).

## Data Availability

The data presented in this study are available on request from the corresponding author. The data are not publicly available due to privacy.

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
