# Peer review of "The Reciprocal Relationship between Frailty and Depressive Symptoms among Older Adults in Rural China: A Cross-Lag Analysis"

_healthcare, 2021, doi:10.3390/healthcare9050593_

Round 1
Reviewer 1 Report
The paper offers a valuable panel analysis related frailty and depression over time. This analysis shows weak cross-lagged effects and stronger stability effects. Given these, the work can have some enhancements to consolidate knowledge building, as suggested as follows.
- Theory relating frailty and depression needs to be stronger and convincing to underpin the analysis. It needs to be prominent as an introduction, rather than a brief note in discussion. The theory essentially needs to formal, established, logical, and succinct rather than nebulous in need of wordy clarification. Review of empirical research does not mean a theory.
- Hypothesis 3 is weak presently and thus unnecessary. Examination of gender invariance does not require a hypothesis.
- It needs to clarify the scoring of frailty. Why were relationships between frailty and depression negative? If they were negative, then the factor was health instead of frailty.
- A preferable alternative to the cross-lagged model is the reciprocal-effect model such that frailty and depression affect each other contemporaneously, given the effects of their prior states. Herein, reciprocal effects, not correlations, are intriguing.
- All analyses need to control for background characteristics, whose effects are noteworthy as well.
- It needs to clarify if the effects are standardized ones (beta instead of b).
- Use of the right words and grammar are important. Proficiency in academic English needs to be demonstrable. Possibly, in case of uncertainly in word meaning, using simple words is enough.
- Articulation the merit of contributing to knowledge advancement is necessary. Arguably, relationships between frailty and depression may be too obvious to be worth a study.
- Self-explanatory variable names, instead of acronyms, are preferable, just given ample space for presentation.
In all, enriching the present work for more solid and justifiable knowledge advancement is necessary.
Author Response
Q1: Theory relating frailty and depression needs to be stronger and convincing to underpin the analysis. It needs to be prominent as an introduction, rather than a brief note in discussion. The theory essentially needs to formal, established, logical, and succinct rather than nebulous in need of wordy clarification. Review of empirical research does not mean a theory.
Response: Thank you for this suggestion. In the existing literature, we have not found a theory that can explain the relationship between frailty and depression, but we have tried to provide an explanation on the relationship between the two in the revised paper.
Q2: Hypothesis 3 is weak presently and thus unnecessary. Examination of gender invariance does not require a hypothesis.
Response: In the revised version of the paper, we deleted the Hypothesis 3.
Q3: It needs to clarify the scoring of frailty. Why were relationships between frailty and depression negative? If they were negative, then the factor was health instead of frailty.
Response: Because higher scores indicate lower level of depression, so relationships between frailty and depression were negative.
Q4: A preferable alternative to the cross-lagged model is the reciprocal-effect model such that frailty and depression affect each other contemporaneously, given the effects of their prior states. Herein, reciprocal effects, not correlations, are intriguing.
Response: Thank you very much for this suggestion. In fact, many studies have shown that frailty and depression coexist among older adults. In this study, we use cross-lagged model to examine the causal relationship between frailty and depression.
Q5: All analyses need to control for background characteristics, whose effects are noteworthy as well.
Response: In the model analysis, we have controlled some sociodemographic variables, such as age, gender, marital status and education.
Q6: It needs to clarify if the effects are standardized ones (beta instead of b).
Response: Yes, the effects are standardized, we have changed b to β in the revised paper.
Q7: Use of the right words and grammar are important. Proficiency in academic English needs to be demonstrable. Possibly, in case of uncertainly in word meaning, using simple words is enough.
Response: Thank you very much for this suggestion. With the help of professionals, we have corrected wrong words and grammar in the revised paper.
Q8: Articulation the merit of contributing to knowledge advancement is necessary. Arguably, relationships between frailty and depression may be too obvious to be worth a study.
Response: The important academic value of this study is to examine the relationship between frailty and depression among older adults in rural China.
Q9: Self-explanatory variable names, instead of acronyms, are preferable, just given ample space for presentation.
Response: Thank you very much for this suggestion. We have used full name instead of acronyms in the revised paper.

Reviewer 2 Report
Overall, content and writing was superior.
Commets:
Introduction Line 27( Reference to statement -evidence) Does this require reference?
Line 41-studies "pointed out" , English language , perhaps "demonstrated "
Line 53 -"show" ( English ) language another phrase perhaps(?).
Line 60 Existing studies show....references to these existing studies .Line 190-210 very descriptive ,great diagram and connection to results.
Conclusion, great with suggestions/ recommendations for future study may be added as well.
Author Response
Q1: Introduction Line 27 (Reference to statement -evidence) Does this require reference?
Response: This statement has become a consensus in the field of public, so it doesn’t require reference.
Q2: Line 41-studies "pointed out", English language, perhaps "demonstrated "
Response: Thank you for this suggestion, we have corrected it in the revised paper.
Q3: Line 53 -"show" (English) language another phrase perhaps(?).
Response: we replaced “show” with “find” in the revised paper.
Q4: Line 60 Existing studies show....references to these existing studies .Line 190-210 very descriptive ,great diagram and connection to results.
Response: We have added references on line 60 in the revised paper.
Q5: Conclusion, great with suggestions/ recommendations for future study may be added as well.
Response: Thank you for this suggestion, we added some recommendations in the part of conclusion.

Reviewer 3 Report
To investigate the reciprocal relationship between frailty and depressive symptoms using longitudinal data among older adults in China., the author used data derived from 2014 and 2017 waves of a longitudinal study of 1367 older adults aged 70-84 years, living in rural areas of Jiangsu Province, China, finding, frailty was a significant predictor of increased depressive symptoms. There are some major concerns in the study, although the theme of the reciprocal relationship between frailty and depressive symptoms is interesting.
- The period of the follow up is too short to determine the causal relationship between frailty and depressive symptoms. The author should discuss about the validity of the cross-lagged panel model and a multiple group model with more caution.
- The age 70-84 years of the subjects is too high to assess the causal relationship between frailty and depressive symptoms. The author should discuss about the validity of the age in the subjects with more caution.
- The author deleted the participants who died before the third wave (N =149). The number of the death must be assessed using the comparison between the death rate between the subjects and the general population living in the subjects, because the author must guarantee the representative in the subjects.
- The author examined cognitive function, using the Mini-Mental Status Examination (MMSE) at two waves. Did the author assess the MMSE in the subjects with mild or severe dementia?
- The test using the cross-lagged panel model showed the absence of bi-directional relationship between frailty and depressive symptoms. Why did the results lead to the first hypothesis that frailty is a predictor of depressive symptoms?
- Although frailty was associated with subsequent increase in depressive symptoms, such that participants with higher levels of frailty increase the risks of depressive symptoms (b = -.088, p < .05), depressive symptoms were not significant predictors of increased frailty (b = -.046, p =.404). The value of the prediction of the depressive symptoms seemed to be much lower as compared to that of the frailty. Are there any rooms of evaluating the value of the prediction using other models?
Author Response
Q1: The period of the follow up is too short to determine the causal relationship between frailty and depressive symptoms. The author should discuss about the validity of the cross-lagged panel model and a multiple group model with more caution.
Response: Thank you for this suggestion. Age range of sample is 70-84 years old in our study, and they live in rural areas with relatively poor living conditions and medical and health resources. If they suffer from frailty or depressive symptoms, they are more likely to get worse in a short time. Therefore, a 3-year period is reasonable for observing frailty and depression changes of sample in this study.
Q2: The age 70-84 years of the subjects is too high to assess the causal relationship between frailty and depressive symptoms. The author should discuss about the validity of the age in the subjects with more caution.
Response: Thank you for this suggestion. In the discussion section, we added an explanation on this suggestion.
Q3: The author deleted the participants who died before the third wave (N =149). The number of the death must be assessed using the comparison between the death rate between the subjects and the general population living in the subjects, because the author must guarantee the representative in the subjects.
Response: Thank you for this suggestion. In this study, as of December 31, 2017 (wave 3 follow-up survey ended), among the 1788 older adults in the baseline survey (149 deaths), a total of 513.2 person-years were observed, an average of 2.86 person-years were observed, and the death rate during the 3-year follow-up period was 2.9%. According to the China’s 2010 Census data, the death rate for the 70-74 age group is 3.06%, 75-79 age group is 4.95%, 80-84 age group is 8.48%, and the death rated for the 70-84 age group is 3.4%. Due to changes in the internal structure of older population, we can infer that the mortality rate of 70-84 years old in this study is close to the national level. Therefore, deleting the death sample can still be representative.
Q4: The author examined cognitive function, using the Mini-Mental Status Examination (MMSE) at two waves. Did the author assess the MMSE in the subjects with mild or severe dementia?
Response: Thank you for this suggestion. We did not assess the MMSE in the subjects with severe dementia. Because the survey data used in this study cannot be answered by others, older adults with severe dementia are not the subjects of this study.
Q5: The test using the cross-lagged panel model showed the absence of bi-directional relationship between frailty and depressive symptoms. Why did the results lead to the first hypothesis that frailty is a predictor of depressive symptoms?
Response: Thank you for this suggestion. The cross-lagged correlation design is to obtain the correlation coefficient between the variable itself and the variable with time, and then determine causal variable and result variable according to the coefficient. When comparing correlation coefficients, there is a basic assumption. If variable A causes a change in variable B, that is, A is the cause variable and B is the result variable. Then the correlation between A1 and B2 should be greater than the correlation between B1 and A2. At the same time, because of the relative stability of the cause variable, the correlation between A1 and A2 will be greater than the correlation between B1 and B2. Therefore, frailty is a predictor of depressive symptoms in this study, which supported hypothesis 1.
Q6: Although frailty was associated with subsequent increase in depressive symptoms, such that participants with higher levels of frailty increase the risks of depressive symptoms (b = -.088, p < .05), depressive symptoms were not significant predictors of increased frailty (b = -.046, p =.404). The value of the prediction of the depressive symptoms seemed to be much lower as compared to that of the frailty. Are there any rooms of evaluating the value of the prediction using other models?
Response: Thank you for this suggestion. Indeed, the results of this study show that frailty is more predictive of depressive symptoms. Regarding whether this conclusion is stable, we will use more survey data and other models in future studies to continue to examine the causal relationship between frailty and depressive symptom.
Reviewer 4 Report
Aims and scope: the manuscript falls within the subject area of the Healthcare journal.
Title: the title complies with the requirements of the Healthcare journal.
Abstract: the abstract complies with the editorial recommendations, is quantitatively and qualitatively correct.
Inrtoduction: lines 60-70 need some references.
Materials and Methods: how was the MMSE scale used? – was its result one of the exclusion criteria? Respondents lived in rural areas – this is a significant burden for the study (it was indicated in the limitations).
Results: statistical calculations do not raise any objections – the section is readable and understandable.
Discussion: unfortunately this section is rather poor in volume – despite the available research in this area.
Conclusions: some information from the results section is repeated in this section. It is worth shortening this section.
References: most of the literature was published after 2010 (21) – it should be considered that the bibliography is rather up-to-date. This section contains minor glitches.
General comment: the manuscript has been carefully compiled, and the research problem itself is intriguing. The drawback of the study is the limited section of the discussion.
Author Response
Q1: Introduction: lines 60-70 need some references.
Response: Thank you for this suggestion. We have added some references in the revised paper.
Q2: Materials and Methods: how was the MMSE scale used? – was its result one of the exclusion criteria? Respondents lived in rural areas – this is a significant burden for the study (it was indicated in the limitations).
Response: Thank you for this suggestion. In the questionnaire of the RuLAS, there are 11 questions in the MMSE scale, including “What month and day is it today?” “How old are you?” “Where are you?” “Where were you born?” “How long have you lived here?” “When was new China founded?” “How many days in a year?” “Who was the chairman of the People’s Republic of China and who was the prime minister?” “Continuous reduction, 100-7=? Subtract 7=?” “Reverse number 6-8-2 or 3-5-2-9” “Show five items that are not related to each other (such as watches, keys, cigarettes, pens, coins), and then take them away, let the subject recall what they have”. A wrong answer is 0point, and a correct answer is 1-4 points. Since the question design of this scale is relatively easy to understand, it will not be a burden on respondents lived in rural areas.
Q3: Results: statistical calculations do not raise any objections – the section is readable and understandable.
Response: Thank you for the evaluation for this part of the paper.
Q4: Discussion: unfortunately, this section is rather poor in volume – despite the available research in this area.
Response: Thank you for this suggestion. We have added some analysis in the discussion part in the revised paper.
Q5: Conclusions: some information from the results section is repeated in this section. It is worth shortening this section.
Response: Thank you for this suggestion. We have deleted some information in the conclusion the revise paper.
Q6: References: most of the literature was published after 2010 (21) – it should be considered that the bibliography is rather up-to-date. This section contains minor glitches.
Response: Thank you for this suggestion. We have added some references before 2010 in the revised paper.

Round 2
Reviewer 1 Report
Please kindly note that the revision has very little improvement toward my expected direction. Notably, the analysis remains, with no correction of the signs of the effects (i.e., the impossible negative relationships between frailty and depression--score reversal is necessary). The desired analysis is about concurrent reciprocal effects, not correlations.
Regarding the responses, those to Queries 1, 4, 8, and 9 are not satisfactory. As such, the work has no theoretical grounds to stand firmly and advance thoughtful understanding.
Author Response
Response to reviewer 1
Comments and suggestions: Please kindly note that the revision has very little improvement toward my expected direction. Notably, the analysis remains, with no correction of the signs of the effects (i.e., the impossible negative relationships between frailty and depression--score reversal is necessary). The desired analysis is about concurrent reciprocal effects, not correlations.
Response: Thank you very much for this suggestion. In the revised paper, we have made score reversal and make new analysis according to the results.
Comments and suggestions: Regarding the responses, those to Queries 1, 4, 8, and 9 are not satisfactory. As such, the work has no theoretical grounds to stand firmly and advance thoughtful understanding.
Response: Thank you for this suggestion. To some extent, physical health and mental health may be two sides of the same coin. Maybe we can clearly distinguish mind and body, when considering physical health and mental health, the two are usually linked. While the theoretical mechanisms responsible for the comorbidity of frailty and depression remain to be explored, the consistency of comorbidity across diverse study population suggests that bi-directional relationship should not be ignored but instead be an important consideration in study design.
Q1: Theory relating frailty and depression needs to be stronger and convincing to underpin the analysis. It needs to be prominent as an introduction, rather than a brief note in discussion. The theory essentially needs to formal, established, logical, and succinct rather than nebulous in need of wordy clarification. Review of empirical research does not mean a theory.
Response: Thank you for this suggestion. In the existing literature, we have not found a theory that can explain the relationship between frailty and depression, but we have tried to provide an explanation on the relationship between the two in the revised paper.
New Response: In the revise paper, we have added the explanation of the study design.
Q4: A preferable alternative to the cross-lagged model is the reciprocal-effect model such that frailty and depression affect each other contemporaneously, given the effects of their prior states. Herein, reciprocal effects, not correlations, are intriguing.
Response: Thank you very much for this suggestion. In fact, many studies have shown that frailty and depression coexist among older adults. In this study, we use cross-lagged model to examine the causal relationship between frailty and depression.
New Response: According to the rationale, the cross-lagged model is used to make reciprocal-effect.
Q8: Articulation the merit of contributing to knowledge advancement is necessary. Arguably, relationships between frailty and depression may be too obvious to be worth a study.
Response: The important academic value of this study is to examine the relationship between frailty and depression among older adults in rural China.
New Response: Actually, frailty and depression of older adults are ignored in China, especially in rural areas. This study may help to arouse the attention to this issue from academia.
Q9: Self-explanatory variable names, instead of acronyms, are preferable, just given ample space for presentation.
Response: Thank you very much for this suggestion. We have used full name instead of acronyms in the revised paper.

Reviewer 3 Report
The manuscript has been improved well according to the comments.
Author Response
Thank you very much for your affirmation of this research and comments.
Reviewer 4 Report
The authors addressed the comments of the reviewer insufficiently (e.g. discussion section).
Author Response
Comments and suggestions: The authors addressed the comments of the reviewer insufficiently (e.g. discussion section).
Response: Thank you for this suggestion. We have added the content of the discussion section in the revised paper.